# Discrete Object Generation
# with Reversible Inductive Construction

**Ari Seff**
Princeton University
Princeton, NJ
aseff@princeton.edu

**Wenda Zhou**
Columbia University
New York, NY
wz2335@columbia.edu

**Farhan Damani**
Princeton University
Princeton, NJ
fdamani@princeton.edu

**Abigail Doyle**
Princeton University
Princeton, NJ
agdoyle@princeton.edu

**Ryan P. Adams**
Princeton University
Princeton, NJ
rpa@princeton.edu

## Abstract

The success of generative modeling in continuous domains has led to a surge of interest in generating discrete data such as molecules, source code, and graphs. However, construction histories for these discrete objects are typically not unique and so generative models must reason about intractably large spaces in order to learn. Additionally, structured discrete domains are often characterized by strict constraints on what constitutes a valid object and generative models must respect these requirements in order to produce useful novel samples. Here, we present a generative model for discrete objects employing a Markov chain where transitions are restricted to a set of local operations that preserve validity. Building off of generative interpretations of denoising autoencoders, the Markov chain alternates between producing 1) a sequence of corrupted objects that are valid but not from the data distribution, and 2) a learned reconstruction distribution that attempts to fix the corruptions while also preserving validity. This approach constrains the generative model to only produce valid objects, requires the learner to only discover local modifications to the objects, and avoids marginalization over an unknown and potentially large space of construction histories. We evaluate the proposed approach on two highly structured discrete domains, molecules and Laman graphs, and find that it compares favorably to alternative methods at capturing distributional statistics for a host of semantically relevant metrics.

## 1 Introduction

Many applied domains of optimization and design would benefit from accurate generative modeling of structured discrete objects. For example, a generative model of molecular structures may aid drug or material discovery by enabling an inexpensive search for stable molecules with desired properties. Similarly, in computer-aided design (CAD), generative models may allow an engineer to sample new parts or conditionally complete partially-specified geometry. Indeed, recent work has aimed to extend the success of learned generative models in continuous domains, such as images and audio, to discrete data including graphs [38, 25], molecules [14, 21], and program source code [37, 30].

However, discrete domains present particular challenges to generative modeling. Discrete data structures often exhibit non-unique representations, e.g., up to $n!$ equivalent adjacency matrix representations for a graph with $n$ nodes. Models that perform additive construction—incrementally building a graph from scratch [38, 25]—are flexible but face the prospect of reasoning over an

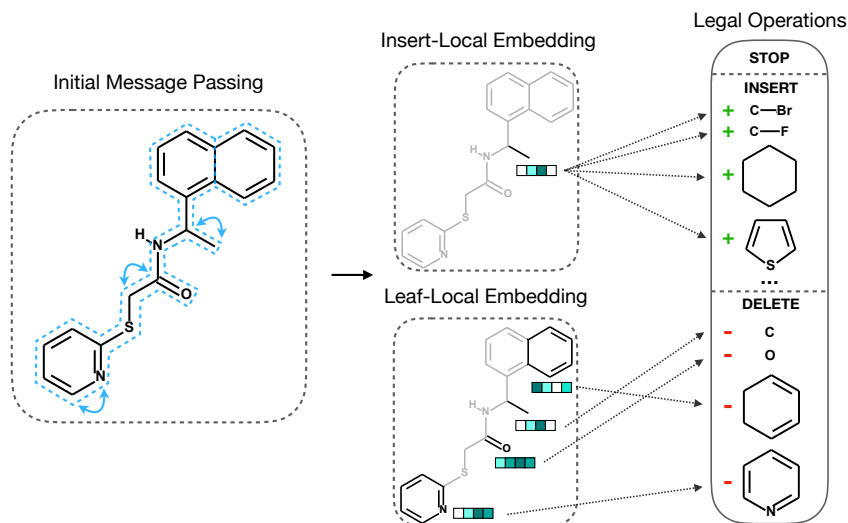

Figure 1: Reconstruction model processing given an input molecule. Location-specific representations computed via message passing are passed through fully-connected layers outputting probabilities for each legal operation.

intractable number of possible construction paths. For example, You et al. [38] leverage a breadth-first-search (BFS) to reduce the number of possible construction sequences, while Simonovsky and Komodakis [34] avoid additive construction and instead directly decode an adjacency matrix from a latent space, at the cost of requiring approximate graph matching to compute reconstruction error.

In addition, discrete domains are often accompanied by prespecified hard constraints denoting what constitutes a *valid* object. For example, molecular structures represented as SMILES strings [36] must follow strict syntactic and semantic rules in order to be decoded to a real compound. Recent work has aimed to improve the validity of generated samples by leveraging the SMILES grammar [21, 7] or encouraging validity via reinforcement learning [18]. Operating directly on chemical graphs, Jin et al. [19] leverage chemical substructures encountered during training to build valid molecular graphs and De Cao and Kipf [8] encourage validity for small molecules via adversarial training. In other graph-structured domains, strict topological constraints may be encountered. For example, Laman graphs [23], a class of geometric constraint graphs, require the relative number of nodes and edges in each subgraph to meet certain conditions in order to represent well-constrained geometry.

In this work we take the broad view that graphs provide a universal abstraction for reasoning about structure and constraints on discrete spaces. This is not a new take on discrete spaces: graph-based representations such as factor graphs [20], error-correcting codes [12], constraint graphs [28], and conditional random fields [22] are all examples of ways that hard and soft constraints are regularly imposed on structured prediction tasks, satisfiability problems, and sets of random variables.

We propose to model discrete objects by constructing a Markov chain where each possible state corresponds to a valid object. Learned transitions are restricted to a set of local *inductive* moves, defined as minimal insert and delete operations that maintain validity. Leveraging the generative model interpretation of denoising autoencoders [2], the chain employed here alternatingly samples from two conditional distributions: a fixed distribution over corrupting sequences and a learned distribution over reconstruction sequences. The equilibrium distribution of the chain serves as the generative model, approximating the target data-generating distribution.

This simple framework allows the learned component—the reconstruction model—to be treated as a standard supervised learning problem for multi-class classification. Each reconstruction step is parameterized as a categorical distribution over adjacent objects, those that are one inductive move away from the input object. Given a local corrupter, the target reconstruction distribution is also local, containing fewer modes and potentially being easier to learn than the full data-generating distribution [2]. In addition, various hard constraints, such as validity rules or requiring the inclusion of a specific substructure, are incorporated naturally.

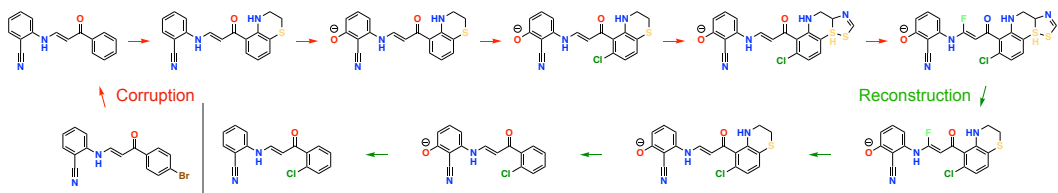

Figure 2: Corruption and subsequent reconstruction of a molecular graph. Our method generates discrete objects by running a Markov chain that alternates between sampling from fixed corruption and learned reconstruction distributions that respect validity constraints.

One limitation of the proposed approach is its expensive sampling procedure, requiring Gibbs sampling at deployment time. Nevertheless, in many areas of engineering and design, it is the downstream tasks following initial proposals that serve as the time bottleneck. For example, in drug design, wet lab experiments and controlled clinical trials are far more time intensive than empirically adequate mixing for the proposed method's Markov chain. In addition, as an implicit generative model, the proposed approach is not equipped to explicitly provide access to predictive probabilities. We compare statistics for a host of semantically meaningful features from sets of generated samples with the corresponding empirical distributions in order to evaluate the model's generative capabilities.

We test the proposed approach on two complex discrete domains: molecules and Laman graphs [23], a class of geometric constraint graphs applied in CAD, robotics, and polymer physics. Quantitative evaluation indicates that the proposed method can effectively model highly structured discrete distributions while adhering to strict validity constraints.

## 2 Reversible Inductive Construction

Let $p(x)$ be an unknown probability mass function over a discrete domain, $D$, from which we have observed data. We assume there are constraints on what constitutes a valid object, where $V \subseteq D$ is the subset of valid objects in $D$, and $\forall x \notin V, p(x) = 0$. For example, in the case of molecular graphs, an invalid object may violate atom-specific valence rules. Our goal is to learn a generative model $p_\theta(x)$, approximating $p(x)$, with support restricted to the valid subset.

We formulate our approach, generative reversible inductive construction (GenRIC)[1], as the equilibrium distribution of a Markov chain that only visits valid objects, without a need for inefficient rejection sampling. The chain's transitions are restricted to legal inductive moves. Here, an inductive move is a local insert or delete operation that, when executed on a valid object, results in another valid object. The Markov kernel then needs to be learned such that its equilibrium distribution approximates $p(x)$ over the valid subspace.

### 2.1 Learning the Markov kernel

The desired Markov kernel is formulated as successive sampling between two conditional distributions, one fixed and one learned, a setup originally proposed to extract the generative model implicit in denoising autoencoders [2]. A single transition of the Markov chain involves first sampling from a fixed corrupting distribution $c(\tilde{x} \mid x)$ and then sampling from a learned reconstruction distribution $p_\theta(x \mid \tilde{x})$. While the corrupter is free to damage $x$, validity constraints are built into both conditional distributions. The joint data-generating distribution over original and corrupted samples is defined as $p(x, \tilde{x}) = c(\tilde{x} \mid x)p(x)$, which is also uniquely defined by the corrupting distribution and the target reconstruction distribution, $p(x \mid \tilde{x})$. We use supervised learning to train a reconstruction distribution model $p_\theta(x \mid \tilde{x})$ to approximate $p(x \mid \tilde{x})$. Together, the corruption and learned reconstruction distributions define a Gibbs sampling procedure that asymptotically samples from marginal $p_\theta(x)$, approximating the data marginal $p(x)$.

Given a reasonable set of conditions on the support of these two conditionals and the consistency of the employed learning algorithm, the learned joint distribution can be shown to be asymptotically consistent over the Markov chain, converging to the true data-generating distribution in the limit of infinite training data and modeling capacity [2]. However, in the more realistic case of estimation

with finite training data and capacity, a valid concern arises regarding the effect of an imperfect reconstruction model on the chain's equilibrium distribution. To this end, Alain et al. [1] adapts a result from perturbation theory [32] for finite state Markov chains to show that as the learned transition matrix becomes arbitrarily close to the target transition matrix, the equilibrium distribution also becomes arbitrarily close to the target joint distribution. For the discrete domains of interest here, we can enforce a finite state space by simply setting a maximum object size.

## 2.2 Sampling training sequences

Let $c(s \mid x)$ be a fixed conditional distribution over a sequence of corrupting operations $s = [s_1, s_2, ..., s_k]$ where $k$ is a random variable representing the total number of steps and each $s_i \in \text{Ind}(\tilde{x}_i)$ where $\text{Ind}(\tilde{x}_i)$ is a set of legal inductive moves for a given $\tilde{x}_i$. The probability of arriving at corrupted sample $\tilde{x}$ from $x$ is

$$c(\tilde{x} \mid x) = \sum_s c(\tilde{x}, s \mid x) = \sum_{s \in S(x, \tilde{x})} c(s \mid x), \tag{1}$$

where $S(x, \tilde{x})$ denotes the set of all corrupting sequences from $x$ to $\tilde{x}$. Thus, the joint data-generating distribution is

$$p(x, s, \tilde{x}) = c(\tilde{x}, s \mid x)p(x) \tag{2}$$

where $c(\tilde{x}, s \mid x) = 0$ if $s \notin S(x, \tilde{x})$.

Given a corrupted sample, we aim to train a reconstruction distribution model $p_\theta(x \mid \tilde{x})$ to maximize the expected conditional probability of recovering the original, uncorrupted sample. Thus, we wish to find the parameters $\theta^*$ that minimize the expected KL divergence between the true $p(x, s \mid \tilde{x})$ and learned $p_\theta(x, s \mid \tilde{x})$,

$$\theta^* = \underset{\theta}{\arg\min} \, \mathbb{E}_{p(x,s,\tilde{x})} \left[ D_{\text{KL}}(p(s, x \mid \tilde{x}) \parallel p_\theta(s, x \mid \tilde{x})) \right], \tag{3}$$

which amounts to maximum likelihood estimation of $p_\theta(s, x \mid \tilde{x})$ and likewise $p_\theta(x \mid \tilde{x})$. The above is an expectation over the joint data-generating distribution, $p(x, s, \tilde{x})$, which we can sample from by drawing a data sample and then conditionally drawing a corruption sequence:

$$x \sim p(x), \quad \tilde{x}, s \sim c(\tilde{x}, s \mid x). \tag{4}$$

## 2.3 Fixed corrupter

In general, we are afforded flexibility when selecting a corruption distribution, given certain conditions for ergodicity are met. We implement a simple fixed distribution over corrupting sequences approximately following these steps: 1) Sample a number of moves $k$ from a geometric distribution. 2) For each move, sample a move type from {Insert, Delete}. 3) Sample from among the legal operations available for the given move type. We make minor adjustments to the weighting of available operations for specific domains. See Appendix F for full details.

The geometric distribution over corruption sequence length ensures exponentially decreasing support with edit distance, and likewise the support of the target reconstruction distribution is local to the conditioned corrupted object. The globally non-zero (yet exponentially decreasing) support of both the corruption and reconstruction distributions trivially satisfy the conditions required in Corollary A2 from Alain et al. [1] for the chain defined by the corresponding Gibbs sampler to be ergodic. Alternatively, one could employ conditional distributions with truncated support after some edit distance and still satisfy ergodicity conditions via the stronger Corollary A3 from Alain et al. [1].

Unless otherwise stated, the results reported in Sections 3 and 4, use a geometric distribution with five expected steps for the corruption sequence length. In general, we observe shorter corruption lengths lead to better samples, though we did not seek to specially optimize this hyperparameter for generation quality. See Appendix A for some results with other step lengths.

## 2.4 Reconstruction distribution

A sequence of corrupting operations $s = [s_1, s_2, ..., s_k]$ corresponds to a sequence of visited corrupted objects $[\tilde{x}_1, \tilde{x}_2, ..., \tilde{x}_k]$ after execution on an initial sample $x$. We enforce the corrupter to be Markov

such that its distribution over the next corruption operation to perform depends only on the current object. Likewise, the target reconstruction distribution is then also Markov, and we factorize the learned reconstruction sequence model as the product of memoryless transitions culminating with a stop token:

$$p_\theta(s_{\text{rev}} \mid \tilde{x}) = p_\theta(\text{stop} \mid x)p_\theta(x \mid \tilde{x}_1) \prod_{i=1}^{k-1} p_\theta(\tilde{x}_i \mid \tilde{x}_{i+1}) \qquad (5)$$

where $s_{\text{rev}} = [s_{k_{\text{rev}}}, s_{k-1_{\text{rev}}}, ..., s_{1_{\text{rev}}}, \text{stop}]$, the reverse of the corrupting operation sequence. If a stop token is sampled from the model, reconstruction ceases and the next corruption sequence begins. For the molecule model, an additional "revisit" stop criterion is also used: the reconstruction ceases when a molecule is revisited (see Appendix D.1 for details).

For each individual step, the reconstruction model outputs a large conditional categorical distribution over $\text{Ind}(\tilde{x})$, the set of legal modification operations that can be performed on an input $\tilde{x}$. We describe the general architecture employed and include domain-specific details in Sections 3 and 4.

Any operation in $\text{Ind}(\tilde{x})$ may be defined in a general sense by a *location* on the object $\tilde{x}$ where the operation is performed and a *vocabulary* element describing which vocabulary item (if any) is involved (Fig. 1). The prespecified vocabulary consists of domain-specific substructures, a subset of which may be legally inserted or deleted from a given object. The model induces a distribution over all legal operations (which may be described as a subset of the Cartesian product of the locations and vocabulary elements) by computing location embeddings for an object and comparing those to learned embeddings for each vocabulary element.

For the graph-structured domains explored here, location embeddings are generated using a message passing neural network structure similar to Duvenaud et al. [9], Gilmer et al. [13] (see Appendix C). In parallel, the set of vocabulary elements is also given a learned embedding vector. The unnormalized log-probability for a given modification is then obtained by computing the dot product of the embedding of the location where the modification is performed and the embedding of the vocabulary element involved. For most objects from the molecule and Laman graph domains, this defines a distribution over a discrete set of operations with cardinality in the tens of thousands.

We note that although our model induces a distribution over a large discrete set, it does not do so through a traditional fully-connected softmax layer. Indeed, the action space of the model is heavily factorized, ensuring that the computation is efficient. The factorization is present at two levels: the actions are separated into broad categories (e.g., insert at atom, insert at bond, delete, for molecules), that do not interact except through the normalization. Additionally, actions are further factorized through a location component and vocabulary component, that only interact through a dot product, further simplifying the model.

## 3   Application: Molecules

Molecular structures can be defined by graphs where nodes represent individual atoms and edges represent bonds. In order for such graphs to be considered valid molecular structures by standard chemical informatics toolkits (e.g., RDKit [24]), certain constraints must be satisfied. For example, aromatic bonds can only exist within aromatic rings, and an atom can only engage in as many bonds as permitted by its valence. By restricting the corruption and reconstruction operations to a set of modifications that respect these rules, we ensure that the resulting Markov chain will only visit valid molecular graphs.

### 3.1   Legal operations

When altering one valid molecular graph into another, we restrict the set of possible modifications to the insertion and deletion of valid substructures. The vocabulary of substructures consists of non-ring bonds, simple rings, and bridged compounds (simple rings with more than two shared atoms) present in training data. This is the same type of vocabulary proposed in Jin et al. [19]. The legal insertion and deletion operations are set as follows:

**Insertion**   For each atom and bond of a molecular graph, we determine the subset of the vocabulary that would be chemically compatible for attachment. Then, for each compatible vocabulary substructure, the possible assemblies of it with the atom or bond of interest are enumerated (keeping its

already-connected neighbors fixed). For example, when inserting a ring from the vocabulary via one of its bonds, there is often more than one valid bond to select from. Here, we only specify the 2D configuration of the molecular graph and do not account for stereochemistry.

**Deletion**  We define the leaves of a molecule to be those substructures that can be removed while the rest of the molecular graph remains connected. Here, the set of leaves consists of either non-ring bonds, rings, or bridged compounds whose neighbors have a non-zero atom intersection. The set of possible deletions is fully specified by the set of leaf substructures. To perform a deletion, a leaf is selected and the atoms whose bonds are fully contained within the leaf node substructure are removed from the graph.

These two minimal operations provide enough support for the resulting Markov chain to be ergodic within the set of all valid molecular graphs constructible via the extracted vocabulary. As Jin et al. [19] find, although an arbitrary molecule may not be reachable, empirically the finite vocabulary provides broad coverage over organic molecules. Further details on the location and vocabulary representations for each possible operation are given in the appendix.

## 3.2 Data

For molecules we test the proposed approach on the ZINC dataset, which contains about 250K drug-like molecules from the ZINC database [35]. The model is trained on 220K molecules according to the same train/test split as in Jin et al. [19], Kusner et al. [21].

## 3.3 Distributional statistics

While predictive probabilities are not available from the implicit generative model, we can perform posterior predictive checks on various semantically relevant metrics to compare our model's learned distribution to the data distribution. Here, we leverage three commonly used quantities when assessing drug molecules: the *quantitative estimate of drug-likeness* (QED) score (between 0 and 1) [4], the *synthetic accessibility* (SA) score (between 1 and 10) [11], and the *log octanol-water partition coefficient* (logP) [6]. For QED, a higher value indicates a molecule is more likely to be drug-like, while for SA, a lower value indicates a molecule is more likely to be easily synthesizable. logP measures the hydrophobicity of a molecule, with a higher value indicating more hydrophobic. Together these metrics take into account a wide array molecular features (ring count, charge, etc.), allowing for an aggregate comparison of distributional statistics.

Our goal is not to optimize these statistics but to evaluate the quality of our generative model by comparing the distribution that our model implies over these quantities to those in the original data. A good generative model would have novel molecules but those molecules would have similar aggregate statistics to real compounds. In Fig. 3, we display Gaussian kernel density estimates (KDE) of the above metrics for generated sets of molecules from seven baseline methods, in addition to our own (see Appendix D for chain sampling details). A normalized histogram of the ZINC training distribution is shown for visual comparison. For each method, we obtain 20K samples either by running pre-trained models [19, 14, 21], by accessing pre-sampled sets [26, 34, 25], or by training models from scratch [33][2]. Only novel molecules (those not appearing in the ZINC training set) are included in the metric computation, to avoid rewarding memorization of the training data. In addition, Table 1 displays bootstrapped Kolmogorov–Smirnov (KS) distances between the samples for each method and the ZINC training set.

Our method is capable of generating novel molecules that have statistics closely matched to the empirical QED and logP distributions. The SA distribution seems to be more challenging, although we still report lower mean KS distance than some recent methods. Because we allow the corrupter to uniformly select from the vocabulary, even if a particular vocabulary element occurs very rarely in training data, it can sometimes introduce molecules without an accessible synthetic route that the reconstructor does not immediately recover from. One could alter the corrupter and have it favor commonly appearing vocabulary items to mitigate this. We also note that our approach lends itself to Markov chain transitions reflecting known (or learned) chemical reactions.

Interestingly, the SMILES-based LSTM model [33] is effective at matching the ZINC dataset statistics, producing a substantially better-matched SA distribution than the other methods. However, as noted in

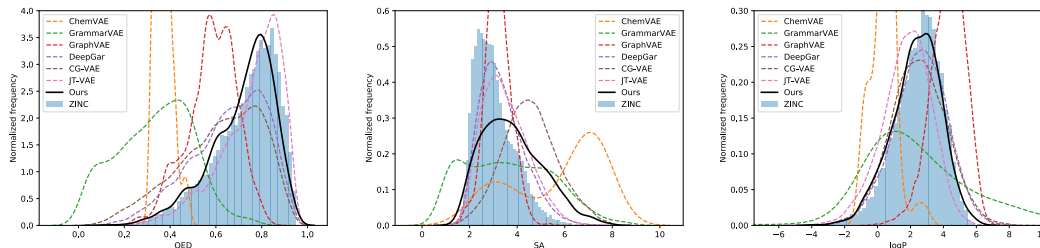

Figure 3: Distributions of QED (left), SA (middle), and logP (right) for sampled molecules and ZINC.

| Source | QED KS | SA KS | logP KS | % valid |
|---|---|---|---|---|
| ChemVAE [14] | 1.00 (0.00) | 1.00 (0.00) | 1.00 (0.00) | 0.7 |
| GrammarVAE [21] | 0.94 (0.00) | 0.95 (0.00) | 0.95 (0.00) | 7.2 |
| GraphVAE [34] | 0.52 (0.00) | 0.23 (0.00) | 0.54 (0.00) | 13.5 |
| DeepGAR [25] | 0.20 (0.00) | 0.15 (0.00) | 0.062 (0.002) | 89.2 |
| SMILES LSTM [33] | 0.022 (0.003) | 0.051 (0.004) | 0.052 (0.004) | 96.1 |
| JT-VAE [19] | 0.090 (0.003) | 0.21 (0.00) | 0.20 (0.00) | 100 |
| CG-VAE [26] | 0.27 (0.00) | 0.56 (0.00) | 0.064 (0.002) | 100 |
| GenRIC | 0.045 (0.003) | 0.28 (0.00) | 0.057 (0.002) | 100 |

Table 1: Molecular property distributional statistics. For each source, 20K molecules are sampled and compared to the ZINC dataset. For SA, QED, and logP, we compute the two-sample Kolmogorov-Smirnov statistic (and its bootstrapped standard error) compared to the ZINC dataset. (Lower is better for the KS statistic.) Self-reported validity percentages are also shown (the value for [14] is obtained from [21]).

[26], by operating on the linear SMILES representation, the LSTM has limited ability to incorporate structural constraints, e.g., enforcing the presence of a particular substructure.

In addition to the above metrics, we report a validity score (the percentage of samples that are chemically valid) for each method in Table 1. A sample is considered to be valid if it can be successfully parsed by RDKit [24]. The validity scores displayed are the self-reported values from each method. Our method, like Jin et al. [19], Liu et al. [26], enforces valid molecular samples, and the model does not have to learn these constraints. See Appendix G for additional evaluation using the GuacaMol distribution-learning benchmarks [5].

We might also inquire how the reconstructed samples of the Markov chain compare to the corrupted samples. See Fig. 6 in the supplementary material for a comparison. On average, we observe corrupted samples that are less druglike and less synthesizable than their reconstructed counterparts. In particular, the output reconstructed molecule has a 21% higher QED relative to the input corrupted molecule on average. Running the corrupter repeatedly (with no reconstruction) leads to samples that severely diverge from the data distribution.

## 4 Application: Laman Graphs

Geometric constraint graphs are widely employed in CAD, molecular modeling, and robotics. They consist of nodes that represent geometric primitives (e.g., points, lines) and edges that represent geometric constraints between primitives (e.g., specifying perpendicularity between two lines). To allow for easy editing and change propagation, best practices in parametric CAD encourage keeping a part well-constrained at all stages of design [3]. A useful generative model over CAD models should ideally be restricted to sampling well-constrained geometry.

Laman graphs describe two-dimensional geometry where the primitives have two degrees of freedom and the edges restrict one degree of freedom (e.g., a system of rods and joints) [23]. Representing minimally rigid systems, Laman graphs have the property that if any single edge is removed, the

| Source | DoD KS | % valid |
|---|---|---|
| E-R [10] | 0.95 (0.03) | 0.08 (0.02) |
| GraphRNN [38] | 0.96 (0.00) | 0.15 (0.03) |
| GenRIC | 0.33 (0.01) | 100 (0.00) |

Table 2: Laman graph distributional statistics. The mean and, in parentheses, the standard deviation, of the bootstrapped KS distance between the DoD distribution for each set of sampled graphs and the training data are shown. In addition, we display mean and standard deviations for bootstrapped validity scores.

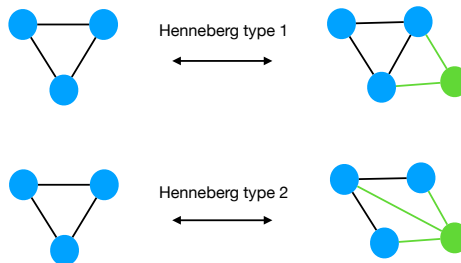

Figure 4: The legal inductive moves for Laman graphs, derived from Henneberg construction [16].

system becomes under-constrained. For a graph with $n$ nodes to be a valid Laman graph, the following two simple conditions are necessary and sufficient: 1) the graph must have exactly $2n - 3$ edges, and 2) each node-induced subgraph of $k$ nodes can have no more than $2k - 3$ edges. Together, these conditions ensure that all structural degrees of freedom are removed (given that the constraints are all independent), leaving one rotational and two translational degrees of freedom. In 3D, although the corresponding Laman conditions are no longer sufficient, they remain necessary for well-constrained geometry.

## 4.1 Legal operations

Henneberg [16] describes two types of node-insertion operations, known as *Henneberg moves*, that can be used to inductively construct any Laman graph (Fig. 4). We make these moves and their inverses (the delete versions) available to both the corrupter and reconstruction model. While moves #1 and #2 can always be reversed for any nodes of degree 2 and 3 respectively, a check has to be performed to determine where the missing edge can be inserted for reverse move #2 [15]. Here, we use the $O(n^2)$ Laman satisfaction check described in [17] to determine the set of legal neighbors. At the rigidity transition, it runs in only $O(n^{1.15})$.

## 4.2 Data

For Laman graphs, we generate synthetic graphs randomly via Algorithm 7 from Moussaoui [29], originally proposed for evaluating geometric constraint solvers embedded within CAD programs. This synthetic generator allows us to approximately control a produced graph's degree of decomposability (DoD), a metric which indicates to what extent a Laman graph is composed of well-constrained subgraphs. Such subsystems are encountered in various applications, e.g., they correspond to individual components in a CAD model or rigid substructures in a protein. The degree of decomposability is defined as $\text{DoD} = g/n$, where $g$ is the number of well-constrained, node-induced subgraphs and $n$ is the total number of nodes. We generate 100K graphs each for a low and high decomposability setting (see Appendix E.1 for full details).

## 4.3 Distributional statistics

Table 2 displays statistics for Laman graphs generated by our model as well as by two baseline methods all trained on the low decomposability dataset (we observe similar results in the high decomposability setting). For each method, 20K graphs are sampled. The validity metric is defined the same as for molecules (Section 3.3). In addition, bootstrapped KS distance between the sampled graphs and training data for DoD distribution is shown for each method.

While it is unsurprising that the simple Erdős–Rényi model [10] fails to meet validity requirements ($< 0.1\%$ valid), we see that the recently proposed GraphRNN [38] fails to do much better. While deep graph generative models have proven to be very effective at reproducing a host of graph statistics, Laman graphs represent a particularly strict topological constraint, imposing necessary conditions on every subgraph. Today's flexible graph generative models, while effective at matching local statistics, are ill-equipped to handle this kind of global constraint. By leveraging domain-specific inductive

moves, the proposed method does not have to learn what a valid Laman graph is, and instead learns to match the distributional DoD statistics within the set of valid graphs.

# 5 Conclusion and Future Work

In this work we have proposed a new method for modeling distributions of discrete objects, which consists of training a model to undo a series of local corrupting operations. The key to this method is to build both the corruption and reconstruction steps with support for reversible inductive moves that preserve possibly-complicated validity constraints. Experimental evaluation demonstrates that this simple approach can effectively capture relevant distributional statistics over complex and highly structured domains, including molecules and Laman graphs, while always producing valid structures. One weakness of this approach, however, is that the inductive moves must be identified and specified for each new domain; one direction of future work is to learn these moves from data. In the case of molecules, restricting the Markov chain's transitions to learned chemical reactions could improve the synthesizability of generated samples. Future work can also explore enforcing additional hard constraints besides structural validity. For example, if a particular core structure or scaffold with some desired baseline functionality (e.g., benzodiazepines) should be included in a molecule, chain transitions can be masked to respect this. Coupled with other techniques such as virtual screening, conditional generation may enable efficient searching of candidate drug compounds.

**Acknowledgements**

We would like to thank Wengong Jin, Michael Galvin, Dieterich Lawson, and members of the Princeton Laboratory for Intelligent Probabilistic Systems for valuable discussion and feedback. This work was partially funded by the Alfred P. Sloan Foundation, NSF IIS-1421780, and the DataX Program at Princeton University through support from the Schmidt Futures Foundation. AS was supported by the Department of Defense through the National Defense Science and Engineering Graduate Fellowship (NDSEG) Program. We acknowledge computing resources from Columbia University's Shared Research Computing Facility project, which is supported by NIH Research Facility Improvement Grant 1G20RR030893-01, and associated funds from the New York State Empire State Development, Division of Science Technology and Innovation (NYSTAR) Contract C090171, both awarded April 15, 2010.

## Footnotes

[1] https://github.com/PrincetonLIPS/reversible-inductive-construction

[2]We use the implementation provided by [5] for the SMILES LSTM [33].

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
