[Supplementary Material]

## A  Geometric Distribution for Corrupter

Fig. 3 displays distributions for molecular samples from models trained with varying geometric distributions for the corruption sequence length. As the sequence length increases (and the corruptions become less local), the models produces worse samples. A short average corruption sequence length of one step seems to lead to a better-matched SA distribution, albeit with slower observed mixing for the Markov chain.

Figure 5: Distributions of QED (left), SA (middle), and logP (right) for the ZINC dataset and models trained with varying expected length corruption sequences.

## B  Molecular Reconstruction Model Operations

We describe the representation assigned to each inductive operation. As described in Section 3, each modification is associated with a location (molecule dependent) and an operation type (molecule independent).

1. Stop: a global operation, naturally associated with the entire molecule. The location embedding is produced by embedding the entire molecule.

2. Delete atom leaf: a deletion operation where the deletion target is a single atom. The vocabulary is unique, and the location is associated with a single atom.

3. Delete ring leaf: a deletion operation where the deletion target is a ring or bridged compound. The vocabulary is unique, and the location is associated with a ring.

4. Insert via atom fusion: an insertion operation where the insertion is performed by attaching at an existing atom. The vocabulary is given by all atoms in each molecule of the vocabulary, and the location is associated with a single atom.

5. Insert via bond fusion: an insertion operation where the insertion is performed by attaching at an existing bond. The vocabulary is given by all bonds belonging to rings in each molecule of the vocabulary, and the location is associated with a single bond in a ring.

Embeddings for locations are computed in the following fashion. We follow a message passing architecture similar to Duvenaud et al. [9], Gilmer et al. [13], which produces a message for each bond and for each atom. The atom messages are transformed and pooled to produce the molecule embedding (used for `stop` prediction). Messages for each leaf atom are also transformed to produce embeddings for `delete leaf atom` actions. Messages for each bond in a leaf ring are transformed and pooled to produce embeddings for `delete leaf ring`. Messages for atoms and bonds are transformed to produce embeddings for `insert via atom fusion` and `insert via bond fusion`.

## C  Training Details

In this section we give a brief description of the choices of parameters in training. We refer the reader to the source code[3] for a full description of the model architecture and parameters.

### C.1 Molecule model

Molecules are converted into graphs in a manner identical to the representation used by [19]. The message passing model runs five steps of message passing. An embedding for the molecule is produced by transforming atom-level messages through a two-layer fully connected network, and aggregating the result through an average-pooling and a max-pooling operation (concatenated). For each task-relevant location, an embedding is produced by transforming and pooling the relevant messages, concatenating those with the molecule representation, and transforming with a two-layer fully-connected network. All messages and hidden layers have size 384.

We train each model for 50 epochs, with the Adamax optimizer and a base learning rate of $2 \times 10^{-3}$ at batch size 128. The base learning rate is scaled linearly with the batch size. We also apply a learning rate schedule, dividing the learning rate by 10 after epochs 12, 24 and 36. Additionally, we apply learning rate warm-up by linearly scaling the learning rate from 0 to its base value during the first five epoch. The training is performed with a batch size of 1024, although we did not see any difference with smaller batch sizes (we did notice some issues with larger batches).

### C.2 Laman model

Laman graphs are encoded for the message passing model with a single node degree feature. That feature is encoded with a Fourier encoding of the node degree. The message passing model runs five step of message passing. An embedding for the graph is produced by transforming the node messages with a two-layer fully-connected network, and aggregated using average and max pooling. Location-specific embeddings are produced in the same fashion as in the molecule model.

We train each model for 30 epochs, with the same optimizer settings as in the molecule model. We use a batch size of 256.

All models are trained using a Nvidia Titan X Pascal (12 GB) graphics card.

## D    Sampling Details

Our proposed models require a Markov sampling step. We describe the details below.

For both the molecule and Laman models, we sample from the chain defined by the trained reconstructor by starting from a random object in the training dataset. The chain then alternatively samples sequences from the corrupter and the reconstructor. In both cases, the results reported in the main text use a corrupter that performs an average of 5 moves (with a geometric distribution).

As we sample from a Markov chain, we do not gather i.i.d. samples. In fact, sometimes the reconstructor returns to the same molecule on adjacent transitions due to perfect reconstruction. The results reported here use every sample from the Markov chain without thinning.

Although validity is maintained through the inductive moves, for both the molecule and Laman models, we in fact encode an action space slightly larger than the true set of valid inductive moves (to make the space more regular). When such an invalid action is sampled by the reconstructor, it is ignored, and another sample is taken. In some very rare instances, the reconstructor repeatedly samples invalid actions, in which case the entire transition (including the corruption) is resampled.

For both the molecule and Laman model, a minimal size is set (one leaf for molecules, and three nodes for Laman), to prevent the chain from deleting the entire object (which would cause problems in terms of the representation). For molecules, we also set a maximum size (in terms of the number of atoms), at 25 atoms, although we found values between 25 and 35 to have little effect on the results.

### D.1    Revisit Stop Criterion

In the molecule setting, we make use of an additional stop criterion which is necessary as our model exhibits high precision and does not have access to any recurrent state which would enable it to increase the probability of stopping as the length of the reconstruction sequence increases.

At each reconstruction step, we keep a history of all the molecules visited by the reconstructor so far, and stop the reconstruction process when the output of the reconstruction model already exists in its history.

We interpret this "revisit" as the model implicitly indicating that the obtained molecule is realistic enough (on average) that it is willing to return to it, despite not indicating stop itself due to the high precision of the model.

# E   Dataset Details

## E.1   Laman

As we did not find high-quality real-world datasets for Laman graphs, we considered some synthetic datasets generated by inductively sampling Henneberg moves in a random fashion. More explicitly, each graph in the dataset is generated using Algorithm 7 of [29], reproduced here as Algorithm 1, where the size $n$ and the probability of selecting Henneberg type I moves $p$ are sampled randomly. For all datasets, we sample $n$ from a normal distribution with mean 30 and standard deviation 5. The distribution of $p$ determines the distribution of the degree of decomposability of the graphs in the dataset. We choose the following distributions of $p$ for each dataset: $p \sim \mathcal{U}(0, 0.1)$ for low decomposability and $p \sim \mathcal{U}(0.9, 1)$ for high decomposability.

---

**Algorithm 1:** Procedure for Generating Laman Graph

---

**input**  : $n$, the number of nodes in the graph. $p$, the probability of choosing a move of type I.
**output :** A Laman Graph.
*Initialize from complete graph on three elements*;
$G \leftarrow K_3$ ;
**for** $i \leftarrow 4$ **to** $n$ **do**
    move $\leftarrow$ Sample (Bernoulli($p$));
    **if** move = *0* **then**
        $G \leftarrow$ ApplyRandomTypeI (G) ;
    **else**
        $G \leftarrow$ ApplyRandomTypeII (G) ;
    **end**
**end**

---

# F   Corrupter Details

## F.1   Molecule Corrupter

We use a single fixed corrupter for all molecule models. To corrupt a molecule, we sample a number of corruption steps from a geometric distribution with the given mean, and iteratively apply Algorithm 2 to the molecule. We made no attempt to optimize the corrupter to produce better samples from the generative model or ease the learning process.

---

**Algorithm 2:** Algorithm for single molecule corruption step

---

**input**  : mol: molecule to corrupt
**output :** mol: corrupted molecule
**if** uniform$()$ < 0.5 **then**
    mol $\leftarrow$ DeleteRandomLeaf (mol);
**else**
    atom $\leftarrow$ GetRandomAtom (mol);
    **if** IsInRing *(atom) and* uniform$()$ < 0.25 **then**
        bond $\leftarrow$ GetRandomBondAtAtom (atom);
        mol $\leftarrow$ InsertRandomAtBond (mol, bond);
    **else**
        mol $\leftarrow$ InsertRandomAtAtom (mol, atom);
    **end**
**end**

---

## F.2 Laman Corrupter

We use a single a single corrupter for all our Laman models. For a single corruption sub-step, this corrupter first chooses among the four action types (Henneberg type I and type II, and their inverses) uniformly at random, and then uniformly samples among the valid actions of the chosen type. As with the molecule corrupter, the number of sub-steps is sampled from a geometric distribution with given mean.

## G  GuacaMol Benchmarks

We also evaluate our model on the new GuacaMol distribution-learning benchmarks [5] after training on the ChEMBL dataset [27]. Using the same hyperparameters as for the ZINC model, we obtain validity: 1.0, uniqueness: 0.933, novelty: 0.942, KL divergence: 0.771, and FCD: 0.058 (see [5] for a description of each metric and comparisons with a few other methods). Note that the FCD score [31] is not directly applicable to our model's samples due to inherent autocorrelation in the generated chains. In part, the FCD score assesses diversity of samples compared to the training set by computing the activation covariance of the penultimate layer of ChemNet. The autocorrelation limits the sample diversity but may be addressed by standard techniques for Markov chains such as thinning. Here, we report results using the same sampling framework as for the ZINC model.

## H  Reconstructed vs. Corrupted Samples

In Fig. 6, we display QED and SA score distributions for the reconstructed molecules ($x$) and the corrupted molecules ($\tilde{x}$) visited during Gibbs sampling as well as molecules generated by solely running the corrupter (with no reconstruction). The corruption-only samples severely diverge from the data distribution.

Figure 6: Distributions of QED (left) and SA (right) scores for reconstructed molecules ($x$) and corrupted molecules ($\tilde{x}$) visited during Gibbs sampling as well as molecules generated via corruption-only.

## I  Example Chains

Below, we display three example chains for the molecular model.

Figure 7: Example chain. The molecule is displayed after each transition of the Markov chain.

Figure 8: Example chain. The molecule is displayed every five transitions.

Figure 9: Example chain. The molecule is displayed every ten transitions.

## Footnotes

[3] `https://github.com/PrincetonLIPS/reversible-inductive-construction`