[Reviews · NeurIPS 2019]

Reviewer 1



Overall I thought this was an interesting paper which was well written with some nice ideas. I can see the potential for impact of their work. My main concern is that the authors don't give a good intuition for why their idea works and I have doubts that there might be (1) something special about the ZINC dataset or (2) some important knowledge about chemistry and/or the dataset which was necessary for their results. While the writing was clear I was left with a lot of questions. The most significant ones are marked with **. Gibbs questions: ** What is the initial state for the gibbs sampler? Did it come from the real dataset? Was only one initial state used? Does it change the result if you use different initial conditions? How much burn-in was needed? I tried to find answers for these questions, apologies if I missed them. ** Figure 1 seems disjoint from the paper. It seems to describe the chemical embedding process but it refers to (1) attention and (2) message passing, both of which are never mentioned again. Expert input: I imagine that the results are highly dependent on the corruptor used. This seems like an important part of the paper and I was required to read the appendix to understand your research. I still was left with many questions: Who wrote the corruptor? Were they an expert in Chemistry? Can the authors expand on how the corruptor choice impacts performance? Are there bad corruptors that don't work? Is chemistry special? I find it surprising that the corruptor, which to me looks basically like a random walk along legal moves, provides a good match for the true distribution of chemical molecules. I would imagine that there are common tropes in chemistry which are not visited often. Perhaps the authors will suggest that its the denoising auto-encoder which is learning about these chemical tropes -- but that is especially suprising. ** Can the authors explain their success? Is there something special about the ZINC dataset that makes this possible? Would this, for example, work for generating student code-solutions (see code.org/research)? Baselines: ** What if you just run the corruptor? How well does it do? ** If an expert is able to define generative legal moves, can they also write a generative grammar? ** Would the expert be able to generate the corruptor if they didn't have access to a ground truth dataset? Is the generative process of chemicals markovian in real life? The authors cite denoising auto-encoders as the motivation behind their corruption-reconstruction idea. How important was the Gibbs contribution? If you just applied the denoising auto-encoder idea directly as suggested in [2] perhaps you wouldn't necessarily need Gibbs sampling. Possible confounds: ** How is the frequency of molecules distributed? I imagine some molecules are more common than others -- are they zipfian distributed? If so the results could be dominated by a few common examples. ** What influences from ZINC could have been used to construct the synthetic dataset? Specifically was any data from ZINC looked at by the researchers or shown to the model?

Reviewer 2



- The method itself is well-motivated and generally makes sense. The problem of generating discrete objects is clearly a very challenging topic, and the paper proposes a simple and reasonable solution, which clearly could be of interest to the community. - Authors clearly indicate the main limitations of their approach. - The paper is well-written and is easy to follow.

Reviewer 3



originality: Using the denoising (for some reason my autocorrect always wants me to write "demonising") autoencoder framework to model structured object construction as a markov chain where transitions are local edits is in my opinion well justified, and an interesting alternative to the previously described models. clarity: - I found this paper quite hard to understand (my background is not statistics). For a general ML conference like NeurIPS, I would suggest to write the paper in a more self-contained matter. Going to the preceding, cited work (Bengio et al 2013) was necessary to get a better idea about what the authors did in this paper. quality: + the disadvantage of the model (expensive sampling) is expressed - The authors yet again introduce a another benchmark for molecule generation. This reviewer has now reviewed 15 papers for molecule generation for NeurIPS, ICML, and ICLR. 14 of them introduce new benchmarks, instead of reusing already established ones created by domain experts. The authors should be required to use the established guacamol benchmark for molecule generation: https://github.com/BenevolentAI/guacamol - Machine Learning has made most progress when a single benchmark was used in many papers (Imagenet!!!) - the CGVAE paper (Liu, Allamanis, Brockschmidt, Gaunt, NeurIPS 2018) and the DeepGAR paper indicate that a simple autoregressive LSTM model on SMILES strings can outperform those models (even though the SMILES LSTM is probably the most boring model in the world). It therefore should to be added as a baseline, regardless of the outcome of the results. It may also be interesting to look at the reviews of the Liu et al paper https://media.nips.cc/nipsbooks/nipspapers/paper_files/nips31/reviews/4855.html . An implementation of the SMILESLSTM can be found here: https://github.com/BenevolentAI/guacamol_baselines/tree/master/smiles_lstm_hc - this model was first reported in https://arxiv.org/abs/1701.01329 Questions: Could the authors point out how the model can be used for structured object optimisation, that is finding objects with optimal properties? _______________________ Added after the authors provided their rebuttal: Thanks to the authors for addressing the questions. I have adjusted my score to 8, and would vote for acceptance, under the condition that the authors add the results of the Guacamol benchmark & baselines, regardless of the outcome (as they wrote in the reply), and add a comment in the outlook on how to employ the model for optimisation tasks.

[Author Response · NeurIPS 2019]

We thank the reviewers for the valuable feedback and address specific comments below.

**Clarification of the Gibbs sampling procedure:** It is not the corruption distribution itself that ultimately generates new, realistic objects; rather, it is the repeated application of the corruption and reconstruction distributions in succession. Running the corrupter repeatedly (with no reconstruction) leads to samples that severely diverge from the data distribution (see Appendix G). Intuitively, the corrupter executes a few random modifications to the current object, "nudging" it off the true data manifold (but still respecting the validity constraints). In turn, the learned reconstructor is trained to undo the corruptions, pushing it back to the data manifold. Ergodic theory states that when a Markov chain is constructed by alternatingly sampling from the conditionals corresponding to a joint distribution, e.g., $p(x \mid \tilde{x})$ and $p(\tilde{x} \mid x)$ for the joint $p(x, \tilde{x})$, asymptotic samples from the chain will be from the marginal distributions. In our approach, we leverage this theory (after Bengio et al., 2013) to construct a generative model from the combination of the fixed corrupter and learned reconstructor. We plan to expand Section 2.1 with additional explanation to make the paper more self-contained.

**Model hyperparameters (sampler initialization, corruption distribution):** Due to space constraints, some of the details of the model hyperparameters, including chain initialization, were included in Appendix D. For both molecules and Laman graphs, each chain is initialized with a random sample from the training set. For each of the experiments, 10 chains are run for 2K transitions each, resulting in 20K samples. Although the samples are not i.i.d., no burn-in or thinning is used. We find that the initial state has minimal effect on the resulting chain-specific KS distances. Similarly, we find (see Appendix A) that for a range of corruption distributions, the performance is not substantially affected (although extreme parameters do affect performance).

**Domain-specific legal moves:** The legal inductive moves approximately correspond to a domain-specific graph-rewrite grammar. Defining such moves does require some domain expertise. While a dataset is not strictly required to define the inductive moves, it can help. For the molecular domain, the moves are based on a primitive vocabulary of bonds and rings obtained via tree decomposition of the training set (similar to Jin et al., 2018). For Laman graphs, we leverage the Henneberg operations from rigidity theory without reference to data. The proposed method is applicable to domains where inductive moves can be specified that preserve some notion of validity. For example, for generating source code, grammar parse trees can be perturbed with insertions and deletions of production rules, altering samples while respecting syntactic validity. The more prior knowledge (e.g., hard constraints) that is encoded in the inductive moves, the less the model has to learn. We plan to apply the method to additional domains including constructive solid geometry and program source code in future work.

**Additional experimental results (Guacamol benchmarks, SMILES LSTM):** We ran the new Guacamol distribution-learning benchmarks after training our model on the ChEMBL dataset. Using the same hyperparameters as for the ZINC model, we obtain validity: 1.0, uniqueness: 0.933, novelty: 0.942, KL divergence: 0.771, FCD: 0.058. Note that these are preliminary results and the FCD score is not directly applicable to our samples. This is due to inherent autocorrelation in the generated chains which is not taken into account by the FCD computation. The autocorrelation may be addressed by standard techniques, e.g., thinning, but there was not time to evaluate this for the author response. We plan to include updated Guacamol results in the paper. We also trained a SMILES LSTM (using the referenced implementation) on the ZINC dataset. Bootstrapped mean (and std) KS distances are as follows: QED: 0.022 (0.003), SA: 0.051 (0.004), logP: 0.052 (0.004). The LSTM is effective at matching the ZINC statistics, producing a much better-matched SA distribution than the other methods. However, the LSTM has limited ability to incorporate structural constraints, e.g., enforcing the presence of a particular substructure.

**Application to structured object optimization:** The proposed method naturally lends itself to substructure-conditional generation ("autocomplete"), which is relevant to a host of design and engineering disciplines. For example, many classes of drugs, e.g., benzodiazepines, are defined by the presence of a core chemical substructure with some desired properties. By masking the inductive moves executed during transitions, the Markov chain can respect this hard constraint. Virtual screening then allows these samples to be efficiently searched, with optimal ones serving as candidates for additional testing.

**Clarification of Figure 1:** The message passing referred to in Figure 1 is described further in Appendix B and C. The attention mentioned in Figure 1 refers to the location-specific embeddings computed for the possible insertion and deletion moves. We inadvertently did not specify this and plan to update the main text accordingly.



[Meta-Review · NeurIPS 2019]

This paper introduces another approach for generating discrete structure with known constraints. Cons: There is a lot of related work in this area already, and this approach seems a little ad-hoc. Pros: The authors ran extra benchmarks for the rebuttal, which raises the standard of the experiments to an acceptable level.